# Modeling of Various Spatial Patterns of SARS-CoV-2: The Case of Germany

**DOI:** 10.3390/jcm10071409

**Published:** 2021-04-01

**Authors:** Albina Mościcka, Andrzej Araszkiewicz, Jakub Wabiński, Marta Kuźma, Damian Kiliszek

**Affiliations:** Faculty of Civil Engineering and Geodesy, Military University of Technology, 00908 Warsaw, Poland; albina.moscicka@wat.edu.pl (A.M.); andrzej.araszkiewicz@wat.edu.pl (A.A.); jakub.wabinski@wat.edu.pl (J.W.); damian.kiliszek@wat.edu.pl (D.K.)

**Keywords:** decision-making, geographic information system, spatial analyses, temporal analyses, entropy, concentration, potential model

## Abstract

Among numerous publications about the SARS-CoV-2, many articles present research from the geographic point of view. The cartographic research method used in this area of science can be successfully applied to analyze the spatiotemporal characteristics of the pandemic using limited data and can be useful for a quick and preliminary assessment of the spread of infections. In this paper, research on the spatial differentiation of the structure and homogeneity of the system in which SARS-CoV-2 occurs, as well as spatial concentration of people infected was undertaken. The phenomena were investigated in a period of two infection waves in Germany: in spring and autumn 2020. We applied the potential model, entropy, centrographic method, and Lorenz curve in spatial analysis. The potentials model made it possible to distinguish core regions with a high level of the growth of new infections, along with areas of their impact, and regions with a low level of generation of new infections. The entropy showed the spatial distribution of differentiation of the studied system and the change of these characteristics between spring and autumn. The concentration method allowed for spatial and numerical demonstration of the concentration of infected population in a given area. We wanted to show that it is possible to draw meaningful conclusions about the pandemic characteristics using only basic data about infections, along with proper cartographic methods. The results can be used to designate the zones of the greatest threats, and thus, the areas where the most intense actions should be taken.

## 1. Introduction

The spread of the SARS-CoV-2 coronavirus in 2020 has created many challenges not only for health systems around the world, but also for scientists from various disciplines. Since the World Health Organization (WHO) [1] declared a pandemic on 11 March 2020, numerous publications have been published about the SARS-CoV-2 virus and the COVID-19 disease that it causes. Among them, many articles present research from the geographic point of view, undertaken by scientists from the fields of cartography, spatial analysis, and geographic information systems (GIS). It has been established that the spread of the SARS-CoV-2 virus is attributable to mobility, and also to some extent to weather.

The large impact of human mobility on the spread of COVID-19 has been analyzed in many studies. Air travel was identified by Wang et al. [2] as one of such factors. The authors identified strong positive correlation between locations of core airports in the United States and COVID-19 clustering regions, whereas in China, rail transport is cited as one of the main factors contributing to the transmission of the virus [3]. Research on people’s mobility is based on data from mobile phones [4,5,6] and Google’s Community Mobility Reports for human mobility modeling [7] to predict degrees of disease spread and to evaluate the effectiveness of health policy strategies. Correlation between distribution of the COVID-19 and population emigration from Wuhan was identified by Chen et al. [8]. Studies also show the impact of lockdowns on SARS-CoV-2 spread [9], its mortality, and reveal the relationship between air pollution, public transport networks, and the development of SARS-CoV-2 [10,11], and the impact of administrative restrictions on the SARS-CoV-2 spread was presented by Warren et al. [6]. These studies cover most continents: Africa [12,13], Asia [14], Europe [15], North America and South America [16,17], and Australia [18]. Gao et al. compared how people in different counties and states reacted to the social distancing guidelines [5]. An analysis of infections by state, incorporating inflows and outflows of interstate travelers, was carried out by Chen et al. [19]. Modeling revealed that curbing interstate travel when the disease is already widespread makes little difference. Increased testing capacity, strict social-distancing, and self-quarantine rules are most effective in abating the outbreak.

Among other publications, papers that revealed a correlation between weather (mainly temperature and humidity) and the pandemic growth are worth mentioning [20,21,22,23]. They present the impact of temperature and humidity on the outbreak [22] and mortality of COVID-19 [20], as well as virus spread and seasonality [21,23]. It turned out that despite the correlation, the role of weather itself as contributing factor to the pandemic growth was relatively insignificant in comparison with other factors, such as people’s mobility and population density [24]. 

Advanced virus spread analyses are often based on machine learning. Niu et al. [25] included Google Trends and machine learning algorithms to build prediction models in order to monitor the virus spread in Italy. Pourghasemi et al. [26] performed spatial modeling, risk mapping, and change detection of COVID-19 in Iran using regression modelling and random forest classification. A more general review of artificial intelligence usage on COVID-19 spatio-temporal data in 49 selected research papers was presented by Jayatilaka et al. [27].

Methods used to model SARS-CoV-2 spread using machine learning as well as traditional mathematical modeling of the infections spread through space and time require multiple data inputs such as place of infection, contacts with other people, environmental and climatic factors, and migration. The key point in such research is to collect data from multiple sources, analyze them, and provide spatial information support for decision making [28]. Detailed and up-to-date data are not always available (especially in developing countries), and researchers are still looking for answers to the questions about the interdependence of factors influencing the spread of SARS-CoV-2. At the current stage of knowledge, decisions made based on the results of such studies are subject to a significant degree of uncertainty, which is confirmed by different strategies used to fight the virus in various countries around the world.

Thus, it was assumed that the essential knowledge necessary in SARS‑CoV-2 spread limitation is what we can know very quickly with limited data sources, i.e., data about infected people along with data about people living in a given area. Having only such data, we can analyze the spatial structure of the area where virus infections occur. The aim of our research was to define the spatial differentiation of the structure of occurrence, homogeneity and spatial concentration of people infected with SARS-CoV-2, along with determination of the changes of these properties over time. To date, no studies of SARS-CoV-2 have examined these aspects by methods used in physics [29], mathematical information theory [30] and econometrics [31,32]. In the study, we have used these methods to address the following research questions:What new information can a study of the spatial structure of SARS-CoV-2 occurrence provide?What spatial relations and dependencies exist between the elements of the tested system?What directions of changes over time can be observed in the distribution, structure, homogeneity (or differentiation), and concentration of SARS-CoV-2 infections?What decisions can be made based on the results of the analysis of the spatial structure of SARS-CoV-2 occurrence?

The novelty of our approach is to use only basic data for quick and preliminary assessment of the spread of infections. It can be valuable to make first decisions and indicate the directions of more detailed analyses requiring new data. The results of our analyses may be useful especially in the initial period of the pandemic, when detailed data are lacking. This creates premises for drawing very important conclusions related primarily to quick decision-making on restrictions and threats, both at the central and local level.

## 2. Materials and Methods

### 2.1. Test Area and Data

The area of Germany was adopted as the test area of research. The necessary data are provided by the Robert Koch Institute in Berlin (RKI) [33]. We used data related to the administrative division units of Germany at the second level of detail (Landkreise and Kreisfreien Städte). In the case of Berlin, the data were aggregated to the NUTS 1 level (The Nomenclature of Territorial Units for Statistics classification is a hierarchical system for dividing up the economic territory of the European Union; NUTS 1 means major socio-economic regions). Based on the analysis of the obtained data, weekly periods were distinguished for the first stage of the spring and autumn waves of infections, when an increasing trend of new infections was noted. Finally, the period 1 March 2020–4 April 2020 (marked as “spring”) and 4 October 2020–7 November 2020 (marked as “autumn”) were analyzed. 

Additionally, the research used data on the population in individual units of the country’s administrative division, obtained from the Eurostat database [34].

### 2.2. Analysis of the Spatial Distribution Using the Potential Model

In this study, we used the concept of potential, similar to that used in physics, resulting from the law of gravity to analyze the distribution of virus infections [29]. The potential model makes it possible to quantify the location of the place of the phenomenon under study as a set of relations to the system of all other places where this phenomenon occurs. The study of relations is transferred here to the spatial structure of the system, considered as a functional whole [35,36]. This model makes it possible to assess the intensity of interactions between the examined administrative units (represented by their centroids) regarding a selected variable, considering the location of the units and their mutual distance. A spatial unit has a specific own potential, but thanks to its location in the region and the system of interactions, its potential may be reduced or increased depending on the potential of neighboring units [37]. 

We used the quotient of potentials as a measure of spatial differentiation of the occurrence of infections in a given area: infections (*CP_i_*) and population (*PP_i_*) for a given administrative unit. These two potentials were used to underline that pandemic development opportunities depend on the population. The quotient of potentials was determined from Formulae (1)–(3) [38,39]:(1)$$P_{i}=\frac{CP_{i}}{PP_{i}}$$
where *P_i_*—quotient of potentials in the *i*-th administrative unit, *CP_i_*—the potential of the SARS-CoV-2 infections in the *i*-th administrative unit, and *PP_i_*—the potential of the population of the *i*-th administrative unit.

The starting point for the application of the potential model was the measurement of the spatial distribution of the population in the form of the population potential of each administrative unit. The potential of the population of the *i*-th administrative unit (*PP_i_*) was determined on the basis of Formula (2):(2)$$PP_{i}=p_{i}+\overset{n-1}{\underset{j=1}{\sum }}\frac{p_{j}}{d_{ij}}$$
where *p_i_*—population in the *i*-th administrative unit, *p_j_*—population in *j-*th administrative unit, *d_ij_*—distance between the centroids of administrative units *i* and *j*, and *n*—number of administrative units in the examined system.

The potential of SARS-CoV-2 infections for the *i*-th administrative unit of *CP_i_* was determined on the basis of Formula (3):(3)$$CP_{i}=c_{i}+\overset{n-1}{\underset{j=1}{\sum }}\frac{c_{j}}{d_{ij}}$$
where *c_i_*—number of infections in the *i*-th administrative unit, *c_j_*—number of infections in the *j-*th administrative unit, *d_ij_*—distance between the centroids of administrative units *i* and *j*, and *n*—number of administrative units in the examined system.

In order to show changes in the quotient of potentials over time, they were determined for two time periods: spring and autumn.

The quotient of potentials is presented using a choropleth map, classifying the obtained results into quintile groups, each of which covers 20% of the analyzed areas. It was assumed that the first quintile group with the highest values of the quotient of potentials would cover the so-called core areas, i.e., areas with a high level of the development of new SARS-CoV-2 infections. Core regions are units that have such a high level of infections that they generate new infections themselves, regardless of external influence. The second and third quintile groups are the areas of influence of the core areas, while the fourth and fifth quintile groups are peripheral regions, with a low level of generation of new infections.

### 2.3. Analysis of Spatial Differentiation with the Use of Entropy

To analyze the homogeneity or differentiation of a spatial phenomenon, we used entropy, which is the basic function of mathematical information theory [30]. Entropy is used in cartography in relation to phenomena distributed in geographical space [40,41] to assess its spatial diversity. In this study, we used entropy to determine the differentiation of SARS-CoV-2 infections per 10,000 people indicators at district level.

In order to determine entropy, we adopted a certain simplification in the analyzed system. The infections indicator was divided into 9 sections (*p_max_* = 9) of equal spread, including similar values of the phenomenon, which is equal to the maximum number of neighboring administrative units in the examined system.

In this study, the entropy was determined for individual administrative units, represented by their centroids, using Formulae (4)–(7) [42,43,44]. The entropy (*H_i_*) for the *i*-th administrative unit was determined from Formula (4):(4)$$H_{i}=-\overset{p_{i}}{\underset{s=2}{\sum }}\omega_{is}\;lg_{2}\omega_{is}$$
where *w_is_*—density of section *s* in the *i*-th administrative unit and *p_i_*—number of sections for the *i*-th administrative unit.

The density *w_is_* was determined for each section (*s*) and expressed as (5):(5)$$\omega_{is}=\frac{m_{is}}{k_{i}}$$
where *m_is_*—number of neighbors in section (*s*) of the *i*-th administrative unit and *k_i_*—number of all neighbors of the *i*-th administrative unit.

The value of *H* index falls in the range from 0 to ${lg}_{2}k$ and may vary for different administrative units that have a different number of neighbors. Therefore, at the beginning, administrative units adjacent to only one unit, such as enclaves, were excluded from the analysis so as not to distort the image of entropy with the assumptions made. 

For the given number of sections (*p_max_* = 9 in our case), maximum entropy (*H_max_*) can be determined. It occurs when the maximum number of sections is equal to the maximum number of neighbors (*p* = *k* = 9) and can be expressed as follows (6):(6)$$H_{max}=\;lg_{2}k$$

On the basis of *H_i_* and *H_max_*, we determined the relative entropy [43] (*h_i_*) (7):(7)$$h_{i}=\frac{H_{i}}{H_{max}}=\frac{-\sum_{s=2}^{p_{i}}\omega_{is}\;lg_{2}\omega_{is}}{lg_{2}k}$$

The value of relative entropy varies in the range from *h* = 0, denoting the maximum concentration of the phenomenon in one section, to *h* = 1, determining the maximum dispersion of the phenomenon—equal share of contamination of neighboring units in each of the sections. 

We determined the relative entropy for the spring and autumn wave of infections. The results of the relative entropy analysis are presented on the maps using the isoline method, which made it possible to present the spatial differentiation of the SARS-CoV-2 infection independently from the administrative division.

### 2.4. Determination of the Concentration Zones

We determined the zones of infection concentration with use of the cartographic concentration method, based on the Lorenz concentration method [32]. Its statistical use was modified by Uhorczak [45] for the purposes of cartography and called the mosaic concentration. In this method, the zones of infection concentration were determined using an ordered series of data on the infection density ($D_{i}$) in each administrative division unit, determined as the ratio of the number of infections (*c_i_*) and the total population in a given unit (*p_i_*) (8):(8)$$D_{i}=\frac{c_{i}}{p_{i}}$$

Having an ordered series of data on the density of infections and the absolute number of infections corresponding to each unit, the number of infected persons was summed up until the assumed values of the designated zones were obtained. We decided to group these values into 10% zones, each zone containing 10% of infected population. The results are presented in the form of a mosaic concentration choropleth map.

Based on the concentration zones, we calculated the percentage of the country’s total area that is occupied by each concentration zone of infected population. The results are presented in the form of a Lorenz curve [46]. It describes the uneven distribution of infections in the study area.

Using the Lorenz curve, we determined the concentration coefficient—the so-called Gini coefficient [31]—showing the inequality in the distribution of infections in the study area in numerical terms. The Gini coefficient (*G*) was determined from the Formula (9),
(9)$$G=\frac{a}{a+b}$$
where *a*—the area between the hypotenuse of the unit triangle under consideration and the Lorenz concentration curve and *b*—the area between the legs of the considered unit triangle and the Lorenz concentration curve.

The Gini coefficient assumes values in the range (0,1). The zero value indicates full uniformity of distribution—the Lorenz curve then coincides with the hypotenuse of the triangle. An increase in the coefficient value means an increase in the inequality of distribution. 

Maps of concentration zones, Lorenz curves, and Gini coefficients were determined separately for spring and autumn.

### 2.5. Analysis of the Variability of the Geographic Center

We carried out the analysis of the variability of the geographical location of the infection center using the so-called centrographic method [43,47]. Using this method, we determined the center of gravity of SARS-CoV-2 infections, also known as the focal point. As in the previous analyses, we assumed that the phenomenon is represented based on the number of infections for each administrative unit, represented by its center of gravity (centroid). Knowing the location of these points, as well as the values of infections at these points in different time intervals, the locations of the resultant points in weekly time intervals were determined, i.e., the geographic centers.

We determined the location of the geographic center in each adopted period of time using Formula (10) [43,47]:(10)$$l_{n}=\frac{\sum_{i=1}^{n}\left(c_{i}\lambda_{i}\right)}{\sum_{i=1}^{n}c_{i}};\;\varphi_{n}=\frac{\sum_{i=1}^{n}(c_{i}\varphi_{i}\cos\lambda_{i})}{\sum_{i=1}^{n}(c_{i}\cos\lambda_{i})}$$
where *j_n_*—centroid latitude for *n* administrative units, *l_n_*—centroid longitude for *n* administrative units, *c_i_*—value of the phenomenon (number of infections) of the *i*-th administrative unit, *φ_i_*—latitude of the centroid of the *i*-th administrative unit, *λ_i_*—longitude of the centroid of the *i*-th administrative unit, and *n*—number of administrative units.

The variability of the location of the center in time is presented on the maps using the signature method. 

## 3. Results

The most common way of presenting data of SARS-CoV-2 infections is the choropleth map. In our case we calculated the infections per 10,000 people in each reference unit. For the studied area of Germany, choropleth maps are presented in Figure 1. The districts which were characterized by the highest percentage of infections were marked: Tirchenreuth for spring and Cloppenburg for autumn.

### 3.1. Spatial Distribution of the Quotient of Potentials

The quotient of potentials (*P_i_*), determined on the basis of Formulae (1)–(3) was presented using choropleth maps (Figure 2). In the analyzed period, *P_i_* ranged from 0.00016 to 0.01034 for spring 2020 and from 0.00063 to 0.01164 for autumn 2020.

The choropleth maps show that during the spring, the quotients of potentials were relatively small for most of the country, which means that the potential for infections was much lower than the potential of the population. In spring, the highest quotient of potentials (0.00164–0.01034) occurred mainly in the south of the country. The lowest quotient of potentials (below 0.00051) covers most of the eastern part of Germany—areas with lower population density. 

On the maps, we see units which are core regions with a high level of the growth of new infections, along with areas of their impact, as well as peripheral regions with a low level of generation of new infections. Although the area occupied by individual quintile groups in spring and autumn is very similar, their spatial distribution is different.

The core regions in spring covered 17.35% of the country’s territory—mainly in the south of Germany. In spring, most of Bavaria and Baden Württemberg were in the 1st quintile group. The same group also included Hamburg, Wolfsburg, and Osnabrück (Lower Saxony), Borken, Coesfeld, Münster, Heinsberg, and Aachen (North Rhine-Westphalia), Cochem-Zell (Rhineland-Palatinate), Saarbrücken (Saarland), and Greiz (Thuringia). There are both rural and urban districts. One can also see the areas of influence of the core regions on the map. This included almost the entire remaining parts of Bavaria and Baden Württemberg, as well as the western part of the country located in the 2nd and 3rd quintile groups, concentrated on the outskirts or in the vicinity of core units. In total, they cover 35.42% of the country’s territory (2nd quintile—17.01%, 3rd quintile—18.41%). Administrative units with the lowest quotient of potentials were peripheral units with a low level of generation of new infections. These were mainly districts located in central-eastern Germany, qualified to the 4th and 5th quintiles. Together, they cover 47.23% of the country’s territory (4th quintile—18.55%, 5th quintile—28.68%).

The autumn wave of the pandemic showed a different spatial distribution of the quotient of potentials. Areas with a high quotient (0.00524–0.01164) were more scattered across the country in autumn. Such concentration did not exist in the southern federal states (Bavaria and Baden-Württemberg). The lowest values of the potential ratio were still present in eastern Germany, but here they assumed values up to 0.00226.

In autumn, new core regions were identified in the east: Berlin and four districts in Saxony: Görlitz, Bautzen, Sächsische Schweiz-Osterzgebirge, and Erzgebirgskreis. The number of core regions in the west of the country also expanded, including Cloppenburg, Vechta and Verden (Lower Saxony), and Bremen, as well as some of the Nordrhein-Westfalen. Despite the increase in the number of core regions, their total area decreased to only 13.05% of the total country’s area. This is mainly due to the concentration of infections in regions with a smaller surface area but a higher population density. In autumn, the situation in the southern part of Germany changed. A significant number of the spring core regions were qualified in autumn into the 2nd and 3rd quintile groups. The remaining areas of influence were concentrated in western Germany and in the east of the country, especially in Saxony. The areas of influence in autumn covered an area that equals 37.59% of the country’s total area (2nd quintile—17.58%, 3rd quintile—20.01%), which was slightly bigger than in spring. The peripheral areas were located in the same part of the country as in spring—in its central-eastern part, covering 49.36% of the country’s area (4th fifth—20.05%, 5th quintile—29.31%).

### 3.2. Spatial Differentiation of Entropy

The entropy of the number of infections per 10,000 people (Figure 3) showed the spatial distribution of differentiation of the studied system of administrative units, as well as the change in these characteristics between spring and autumn. 

The spatial distribution of the relative entropy in spring (Figure 3, left map) shows the differentiation of the studied system only in the south of the country. Relative entropy was highest in south and west part of Germany but it exceeded 0.6 only in Schwandorf and Kelheim (both in Bavaria) and in Tuttlingen (Baden-Württemberg). It was also high, reaching 0.5, in the west of the North Rhine-Westphalia. The rest of the country could be described as homogeneous, with the relative entropy not exceeding 0.3, i.e., 72.5% of districts. Of the districts, 37.6%, the less populated areas of north-west Germany, showed zero entropy. This means that many administrative units had a very similar epidemic situation: there were no major differences in the infections between neighboring districts, which was probably due to the small number of infections occurring in most of the country in spring. Only the south of the country, the area of Bavaria and Baden-Württemberg, and states along the border with France were characterized by some variation between administrative units. The infections first appeared mainly in large Bavarian cities, later also affecting rural districts, hence the characteristic increase in the diversity of the system in the south of the country.

In autumn (Figure 3, right map), there was a significant change in the homogeneity of the system. The differentiation of the studied system spread practically over the entire country, showing the chaotic evolution of the epidemic. Almost 70% of administrative units have a relative entropy greater than 0.3 and in over 12% of them it was greater than 0.6 The system of isolines points to a zone from the east part of Lower Saxony to Hesse as region with greatest entropy. Such high values of entropy in practically all of Germany prove the great diversity of the studied system in autumn. Over 50% of districts were represented by relative entropy higher than 0.4. Merely 6% of districts showed zero entropy. Only the border of Brandenburg and Mecklenburg-West Pomerania and the western part of Baden-Württemberg showed a low level of entropy. The rest of the country was already highly diversified.

In autumn the distribution of isolines centered around larger urban centers is common (Figure 3, right map). Densely populated cities like Berlin, Hamburg, and Munich showed less entropy than their neighbors. This means that there were significant differences in the SARS-CoV-2 infections between them and more homogeneous neighboring administrative units. This is mainly due to the existence of many large outbreaks of infections occurring in both larger and smaller cities, and a much lower number of COVID-19 cases in neighboring districts.

### 3.3. Mosaic Concentration Zones

The cartographic method of concentration made it possible to determine the concentration zones of the population infected with SARS-Cov-2. Each zone (where 1 is the highest concentration zone—darkest color in Figure 4), consists of 10% of the total infected population. Choropleth maps of mosaic concentration for spring and autumn are presented in Figure 4. 

Concentration zones with the highest infection concentration in spring are located in the south of the country, mainly including Bavaria and Baden Württemberg, as well as in the western areas of Nordrhein-Westfalen and Rheinland-Pfalz. In spring, 10% of the infected population corresponded to approximately 9700 people. At that time, the first concentration group, including the areas with the highest density of infections, included 18 districts with a total area of about 13,500 km^2^, which accounts for 3.78% of the entire country. The average infection density for the first concentration zone was 0.42% at that time. A total of 40.22% of the country area was covered by units with the lowest infection density (concentration group 10: 123 districts).

In autumn, the zones with the highest infection density were scattered across the country. Only some of them still existed in Bavaria. The structure of the first concentration group changed and included some new districts, mainly in southern Saxony and Niedersachsen-Lower Saxony. In autumn, 10% of the infected population corresponded to approximately 36,900 cases. At that time, the average density of infections for the first zone was 0.91% and it occupied the area of 16 districts, i.e., about 10,300 km^2^, which corresponds to 2.88% of the entire country area. In autumn, the districts with the lowest infection density (128) covered 42.36% of the country’s area.

The differences in the concentration of infected people between spring and autumn are also visible on the Lorenz concentration curve (Figure 5). The green curve representing spring shows a lower concentration of infected population in this period. This is confirmed by the value of the Gini coefficient at the level of 0.422 (42.2%). In autumn, the infections spread throughout the country, but the places of the greatest concentration included mainly major cities and large outbreaks of infections. The Gini coefficient for autumn is 0.525 (52.5%).

The analysis of maps (Figure 4) may at first suggest that the concentration was higher in spring, as the infections were concentrated only in the southern part of the country, while the remaining, a much greater part of the country, was almost unaffected by the virus. However, this is very misleading, as confirmed by a Gini coefficient values. The fact is that spring infections were mainly concentrated in the south of the country, but they covered a large area, almost all of Bavaria and Baden Württemberg. The first three zones, with the highest density of infections, cover 11.50% of the country’s area, while the last two zones, with the lowest density of infections, cover more than half of the country’s area—52.10%. Looking at the autumn map (Figure 4, right map), one can see that there are definitely fewer districts covered with dark colors. This means that a much smaller area was inhabited by a large percentage of infected people. They were concentrated in almost half the area smaller than in spring—the first three zones in autumn occupied only 7.16% of the country’s area. At the same time the area covered by the last two zones (the lowest density of infections) increased to 59.52%. This means that the concentration in autumn increased, which confirms that the pandemic was developing faster in large urban centers, with higher population density, where the virus is transmitted more easily.

### 3.4. Variability of the Location of the Geographical Center

Figure 6 shows the change in the location of geographic center of infections during spring and autumn. During spring, this center moved from the western part of the country (first confirmed cases of COVID-19), it moved west and then south. During autumn, the geographic center of infections moved only slightly, close to the population and the geographic center of the country. The stabilization of the infection center of gravity proves that the pandemic has spread practically all over the country.

## 4. Discussion

The main findings of the study are that the system created by the area with SARS-CoV-2 infections is not uniform, and different kind of relations between administrative units play a significant role in an infection spread.

Results of our research show that infections in one area are influenced not only by internal conditions, but also by infections in neighboring areas. Proposed methods can be used to determine areas where relations between adjacent units have a strong impact on SARS-CoV-2 spread. Our research allowed to designate areas with high infection rate and areas of their influence, because of their economic, social, education, and transport connections. Core regions cannot be developed by chance because core regions depend not only on the pandemic situation in the selected unit, but also its vicinity. A random bad pandemic situation in one unit (i.e., one large center of infections) is not enough to calculate a high potential of infections. Only a group of a few related centers of infections can form a core region. 

Using the proposed methods, the impact areas of the core units can be identified. These areas are at risk of a significant increase in the number of infections. The introduction of restrictions is met with considerable social opposition, but intervention in places where the epidemic has not yet fully developed may allow introducing less severe restrictions. 

The proposed method, as it currently stands, has no validated predictive ability. At the current stage of knowledge about the development of a pandemic, when the bigger amount of data is already available, it is possible to assess it. The assessment could be based on the comparison of the data used in the research with the data from the later periods and checking if the conclusions from the research results are confirmed by the further development of the pandemic. During the assessment, not only the increase/decrease of the number of infections, but also the epidemic policy of the state services should be taken into account. Such assessment will allow to define the usefulness of the method for quickly predicting the directions of pandemic development and designating areas particularly at risk or requiring urgent intervention without detailed data on people mobility, which are not always available and require advanced processing. In future research, we would like to validate the predictive ability of the methods proposed. If validated, the obtained results could be used by public health officials as well as local and state authorities. Quick knowledge about identified risk areas enables efficient response and improves the access to medical equipment and personnel before the situation in a given area becomes very difficult. This is important when planning, for example, the location of temporary hospitals, the number of beds, and the provision of staff for them. In addition, local services, having foreknowledge, could react locally, without the need to restrict large areas, which is of great importance for an economy heavily affected by the pandemic. They can designate the zones of the greatest threats, where the most intense actions should be taken, including, for example, imposing restrictions on social life, and as a result, reducing the impact on human health [48,49]. The knowledge about the structure of the system could allow making decisions appropriate to the situation, i.e., introducing restrictions only in a specific area. It could also help to introduce the same restrictions or the same medical protocols on units with the same nature and structure of the pandemic, not only based on the number of infections. This kind of insight could be used by governments or local authorities to better foresee pandemic development and thus, impose restrictions, the nature of which will depend on the local specificity of the virus spread, instead of introducing nationwide lockdowns. Quick decisions made by authorities are also of high importance for business owners and entrepreneurs as the prior knowledge of the planned restrictions would allow them to manage their businesses in a more controlled way. The possible actions described provide an answer to the fourth research question.

The policy of fighting a pandemic cannot take place only within individual countries. An example is the situation in border regions, clearly visible on practically every map presented. Both in spring and autumn, there are significant outbreaks of infections at the borders. Especially for the German–Czech [50] and German–French border. In spring, large numbers of infections occurred in Bavaria (DE) bordering the Zapadocesky region (CZ), where there were also higher infections in spring. In autumn, significant infections occur in the neighboring regions of Saxonia (DE) and Severocesky (CZ). This clearly shows that the state border is not a barrier to the development of the pandemic. Border regions should work closely together at the local level, both in the fight against the spread of infections and the way in which infected people are controlled. The research methods we propose enable the quick identification of such places, the assessment of the situation and directions of the pandemic development, and consequently, enable efficient decision making. Proposed methods can be used regardless of the quality of the health system, as well as in underdeveloped countries, in the absence of other data.

## 5. Conclusions

This pandemic has been the first such experience in the lives of the present generations, therefore it is important not only to comprehensively understand the very nature of the virus, but also the directions and possibilities of its spatial spread. The year 2020 has shown that the virus epidemic can surprise us a lot and paralyze the lives of most countries in the world for many months. Due to the emerging opinions that we should anticipate more than one such pandemic in the near future, it is necessary to work out ways to quickly investigate and predict the spread of such diseases, in which case the methods proposed in this paper could be used. 

The presented results show the studied phenomenon for selected weeks in spring and autumn 2020. In the near future, we plan to conduct similar studies for individual age groups and sexes to check whether the characteristics of the phenomenon are constant for all infected groups. It will also be interesting to analyze the changes in weekly cycles, which will enable a detailed analysis of the spread and changes in the structure of the infected population.

The proposed research methods and the results obtained from them could be interesting when comparing the manner and pace of epidemic development in different countries. It will also enable the evaluation of the effectiveness of actions taken by different governments. Last but not least, it will be crucial to analyze the changes following the introduction of mass vaccination against COVID-19.

## Figures and Tables

**Figure 1 jcm-10-01409-f001:**
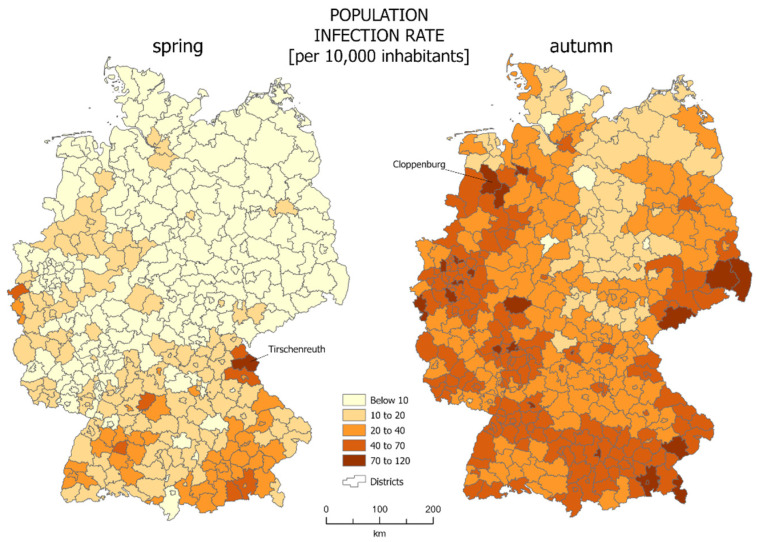
SARS-CoV-2 infections per 10,000 inhabitants in German districts—those with highest infection indicator for both periods are marked (source: own work).

**Figure 2 jcm-10-01409-f002:**
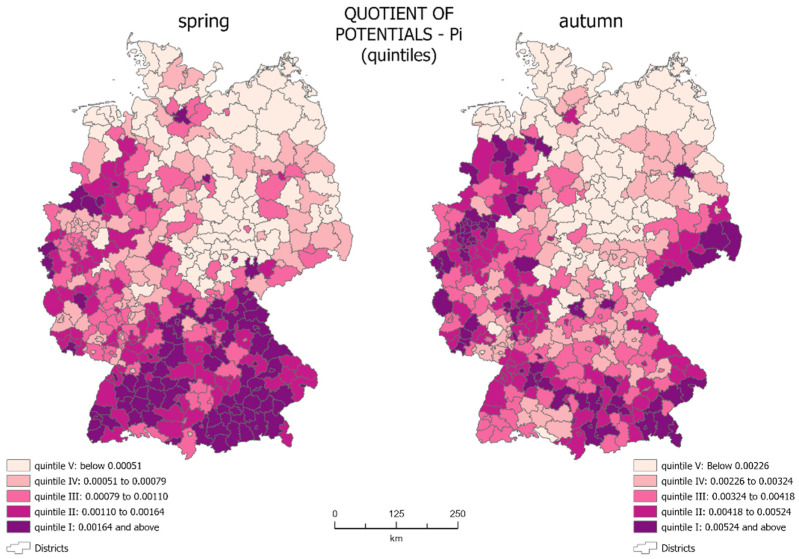
Choropleth maps of quotient of potentials (source: own work).

**Figure 3 jcm-10-01409-f003:**
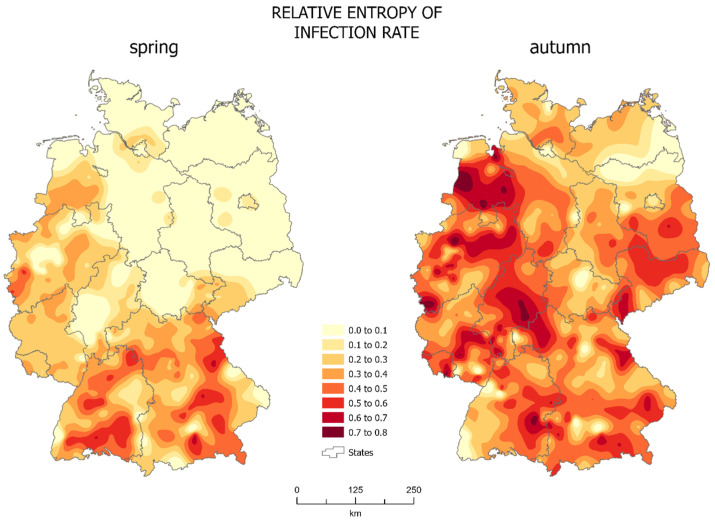
Relative entropy (source: own work).

**Figure 4 jcm-10-01409-f004:**
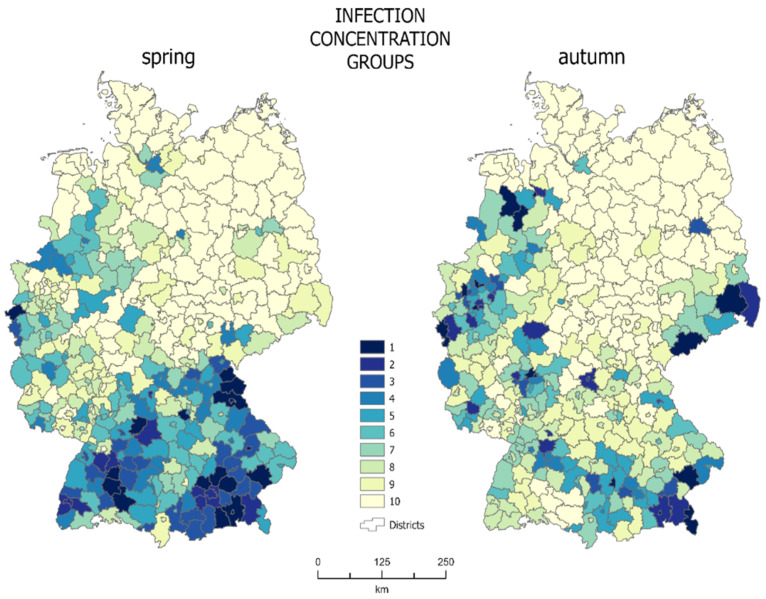
Choropleth map of mosaic concentration zones (source: own work).

**Figure 5 jcm-10-01409-f005:**
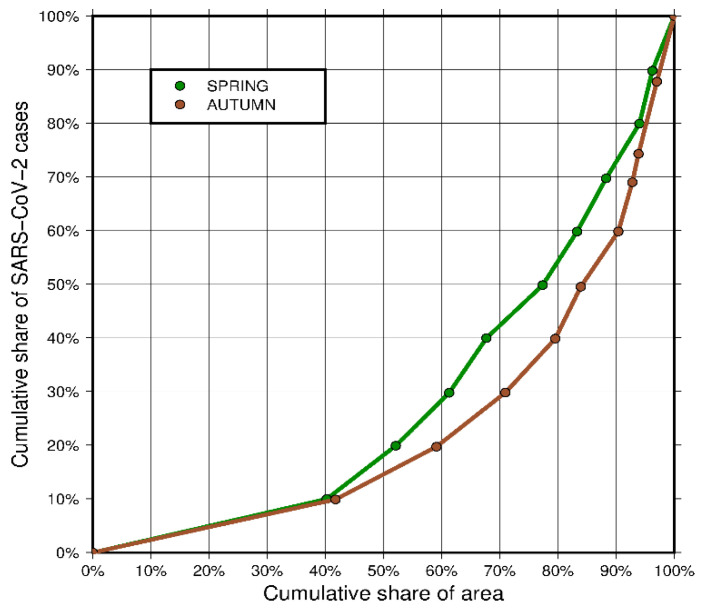
Lorenz concentration curve (source: own work).

**Figure 6 jcm-10-01409-f006:**
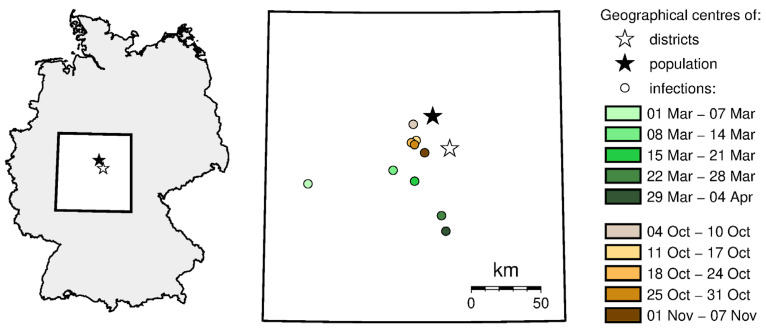
The geographic center of infections movement across the analyzed time (source: own work).

## Data Availability

Publicly available datasets were analyzed in this study. This data can be found here: https://npgeo-corona-npgeo-de.hub.arcgis.com (accessed on 16 February 2021).

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
