# Peer review of "Modeling of Various Spatial Patterns of SARS-CoV-2: The Case of Germany"

_jcm, 2021, doi:10.3390/jcm10071409_

Round 1
Reviewer 1 Report
Modelling of Various Spatial Patterns of SARS-CoV-2: Case of Germany
This is an interesting paper to read. The method introduced is very interesting and promising for future studies. Based on the way I understand the paper, I have the following comments to improve the paper.
- The methodology is interesting and can be applied to other research, such as in traffic engineering. The data collection and the reason for the data to be collected in the two periods make sense.
- The authors have used several metrics that show a similar outcome. It would be beneficial to a reader to have a short but concise paper. For instance, Figure 3 and Figure 4 display the same findings. I find Figure 4 more informative than Figure 3, but the narratives are nearly repetitive.
- Results and discussions need to be together. In most cases, the authors have provided narratives of what they have presented in the figures then discuss them in the next section. For instance, lines 413-417 is describing the difference in the quotient of potential for spring and autumn. The reader needs to move to the next section to understand what do the differences mean. Readers will lose focus.
- One of the importance of the research findings is the applications. This paper lacks a paragraph/section that discusses the application of the findings.
- The language used in this paper is like the authors are responding to a paper by Franch-Pardo et al, 2020. I think the focus should rather be on the introduction of the new methodology while discussing the shortcomings of the current methodologies.
- In the abstract, what does the second sentence mean? To be specific, what does “this” in line 10 mean?
- The introduction/literature review covers a greater number of papers. However, the authors need to show how the proposed approach is stronger than the approaches used in the current studies. Further, machine learning-related pieces of literature that discuss the spatial distribution of COVID19 is not well covered. Most recent studies that cover the spatial distribution of COVID 19 and their associated themes ( Queiroz et al., 2020) using machine learning approaches have not been reviewed.
References
Franch-Pardo, I.; Napoletano, B. M.; Rosete-Verges, F.; Billa, L. Spatial analysis and GIS in the study of COVID-19. A review. Sci Total Environ. 2020, 739, 140033. https://doi.org/10.1016/j.scitotenv.2020.140033.
Queiroz, M. M., Ivanov, D., Dolgui, A., & Fosso Wamba, S. (2020). Impacts of epidemic outbreaks on supply chains: mapping a research agenda amid the COVID-19 pandemic through a structured literature review. Annals of Operations Research, 1–38. https://doi.org/10.1007/s10479-020-03685-7
Author Response
We would like to thank the reviewer for detailed comments and insightful suggestions. They were invaluable for raising the level of our work.
Response to Reviewer 1 Comments
We would like to thank the reviewer for detailed comments and insightful suggestions. They were invaluable for raising the level of our work. Responses to the comments are provided below.
- The methodology is interesting and can be applied to other research, such as in traffic engineering. The data collection and the reason for the data to be collected in the two periods make sense.
Authors’ Reply: We are happy to hear that! Thank you.
- The authors have used several metrics that show a similar outcome. It would be beneficial to a reader to have a short but concise paper. For instance, Figure 3 and Figure 4 display the same findings. I find Figure 4 more informative than Figure 3, but the narratives are nearly repetitive.
Authors’ Reply: Thank you for the suggestion. We have included both maps on purpose to emphasize the influence of the cartographic presentation method on the information conveyed by the map - its feature that is often abused by mapmakers. Infections and the resulting spatial phenomena do not spread only within administrative units, therefore the isoline method (Fig. 4) is important to show the tendencies in spatial distribution of the phenomenon. But both, the data used in the research and decisions related to the government's policy concern individual administrative units. Especially when - as in this case - we want to show core regions, areas of their impact and peripheral regions. In this case the choropleth method is necessary to present the results (Fig. 3).
For a better understanding, we have reworded some paragraphs and added some clarifications in the Discussion (lines: 757-865).
- Results and discussions need to be together. In most cases, the authors have provided narratives of what they have presented in the figures then discuss them in the next section. For instance, lines 413-417 is describing the difference in the quotient of potential for spring and autumn. The reader needs to move to the next section to understand what do the differences mean. Readers will lose focus.
Authors’ Reply: Suggested description of maps and results have been moved from Discussion to Result section.
- One of the importance of the research findings is the applications. This paper lacks a paragraph/section that discusses the application of the findings.
Authors’ Reply: Discussion has been extended. Potential application of the findings was described in detail (lines: 828-865).
- The language used in this paper is like the authors are responding to a paper by Franch-Pardo et al, 2020. I think the focus should rather be on the introduction of the new methodology while discussing the shortcomings of the current methodologies.
Authors’ Reply: According to the reviewer’s suggestion, the Introduction has been redrafted and shortened.
- In the abstract, what does the second sentence mean? To be specific, what does “this” in line 10 mean?
Authors’ Reply: The Abstract has been corrected. Unclear sentences have been redrafted (lines: 10-26).
- The introduction/literature review covers a greater number of papers. However, the authors need to show how the proposed approach is stronger than the approaches used in the current studies.
Authors’ Reply: Both, in the Introduction (generally) and in the Discussion (in detail) we explained the advantages of the approach we used.
- Further, machine learning-related pieces of literature that discuss the spatial distribution of COVID19 is not well covered. Most recent studies that cover the spatial distribution of COVID 19 and their associated themes (Queiroz et al., 2020) using machine learning approaches have not been reviewed.
Authors’ Reply: Machine learning-related literature has been added.
Reviewer 2 Report
The authors undertake the important task of trying to identify whether spatial regions exist where the dynamics of COVID-19 cases is at odds with surrounding areas, or, at least, marched to the beat of a different drummer.
My biggest concern is that this paper buries the lede. At the end of reading the abstract I do not know the central finding of the paper. Nor did I know it after reading the paper full one time. I believe this is an issue of exposition and a lack of biomedical context. I assume the authors want not to demonstrate the rule-of-thumb that infectious diseases start in densely populated areas and diffuse out, but rather to showcase a method.
Poor presentation and a lack of context mar this paper. However, I have only a few methodological concerns. The underlying approach is sound and, with work, I believe a revised version of this manuscript could make a contribution to the field.
Methodological Questions:
- What perspective does Figure 5 provide that Figure 2 does not? If I understand correctly, Figure 5 shows a measure that is a monotonic function of a few adjacent units in FIgure 2. Please describe the additional insight calculating the relative entropy gives.
- I presume SARS-CoV-2 infections means found to have viremia by PCR? Were testing conditions constant throughout this period? In many areas of the world testing standards expanded substantially over the first few months, changing the population being sampled.
- Is there a validation? The authors make the point that their method is easier to calculate for areas without abundant public health data. Germany provides a nice arena to compare it with prior estimates, but I do not believe the authors compare their results with any other methods.
- How does this relate to morbidity and mortality? SARS-CoV-2 would not be a topic of discussion if the virus were less virulent. If this is to be used in areas with less robust public health systems, then it should track numbers with direct relevance to public health
- Could these areas of independence arise from chance? The authors do not provide an analysis of how many regions their method would identify as independent even if applied to, for example, spatially separate centers that were perfectly correlated.
- Many of the hotspots, especially in autumn, hug the borders of Germany. Could this reflect cross-border issues with France, Austria, the Czech Republic? As with (3), the model is more useful to public health officials if it tracks real-world phenomena in an insightful way.
Issues with Presentation:
- Please consider re-writing the introduction and conclusion with an eye to brevity and clarity. I read the Introduction 4 times and still did not take away much more than "Researchers have made data-intensive models. We tried to make do with a more minimal more principled model."
- You never explicitly define what you mean by independence nor comment how this could relate to the administrative structures or local public health measures instituted by those structures.
Author Response
We would like to thank the reviewer for detailed comments and insightful suggestions. They were invaluable for raising the level of our work.
Response to Reviewer 2 Comments
We would like to thank the reviewer for detailed comments and insightful suggestions. They were invaluable for raising the level of our work. Responses to the comments are provided below.
Methodological Questions:
- What perspective does Figure 5 provide that Figure 2 does not? If I understand correctly, Figure 5 shows a measure that is a monotonic function of a few adjacent units in Figure 2. Please describe the additional insight calculating the relative entropy gives.
Authors’ Reply: Figure 2 presents infections per 10 000 inhabitants, it means the density of infections in each single unit – the number of infections in relation to the population. Figure 5 presents the levels of differentiation of the structure of the local system created by the adjacent units. It means that map presents the regions with similar nature of the pandemic on the level higher than single units. These regions are based on the local situation and are defined by taking into account the surroundings of each administrative unit, as the border is not a barrier to the spread of the virus. Diversity of the system (higher entropy) shows that in close neighborhood there are units with different pandemic situation. Homogeneity of the system (lower entropy) shows where in the system is similar pandemic situation. The knowledge about the structure of the system allows to make decisions appropriate to the situation, i.e. introduce restrictions only on a specific area, not on the whole country.
For a better understanding, we have reworded some paragraphs and added some clarifications in the Discussion.
- I presume SARS-CoV-2 infections means found to have viremia by PCR? Were testing conditions constant throughout this period? In many areas of the world testing standards expanded substantially over the first few months, changing the population being sampled.
Authors’ Reply: Thank you for this comment. That is correct and for this reason, we haven’t really focused on absolute values but rather spatial distribution of infections during 2 “separate” waves of pandemic: spring and autumn. Besides, we found the data about number of tests carried out were not fully reliable. The data about tests were aggregated to full weeks. They were often erroneous and only later corrected. We tracked the number of tests carried out and it turns out that it grew dynamically in the first weeks of the pandemic and reached over 1 million tests per week in the middle of year 2020. This statistic remained approximately the same for the rest of the period analyzed.
- Is there a validation? The authors make the point that their method is easier to calculate for areas without abundant public health data. Germany provides a nice arena to compare it with prior estimates, but I do not believe the authors compare their results with any other methods.
Authors’ Reply: The idea of our approach was to use only basic data, i.e. data on virus infections and population and use them for quick and preliminary assessment of the spread of infections. The results obtained in this way can be valuable to make first decisions and indicate the directions of more detailed analyzes requiring new detailed and up-to-date data. We already work on research with the use of health data, but the aim of such study is completely different. Proposed method can be used regardless of the quality of the health system, also in underdeveloped countries, in the absence of other data.
- How does this relate to morbidity and mortality? SARS-CoV-2 would not be a topic of discussion if the virus were less virulent. If this is to be used in areas with less robust public health systems, then it should track numbers with direct relevance to public health.
Authors’ Reply: As the reviewer points out, morbidity and mortality analysis should be conducted in a wider area, considering different health care systems. This is why we did not consider morbidity and mortality, as they are strongly correlated with the standard of living and the quality of health care. This is the plan for the future. Here we propose methods for the analysis of basic data. As we explained in the previous comment, we are currently conducting such research and include a larger number of indicators. We also write about it in the conclusions.
- Could these areas of independence arise from chance? The authors do not provide an analysis of how many regions their method would identify as independent even if applied to, for example, spatially separate centers that were perfectly correlated.
Authors’ Reply: We write in the text that we assumed that units in 1st quantile are the core regions, what means that it is 20% of the units with the highest quotient of potentials. Core regions can’t arise from chance, because the potential of infections - which relation to the potential of population decide of the value of quotients of potentials - depends not only on the pandemic situation in the selected unit, but also on the situation in units related to this one. Random bad pandemic situation in one unit (i.e. one large centre of infections) is not enough to calculate high potential of infections. Only a severe pandemic situation in a few related units (a few centres of infections) can result in high potential of infections, and finally high value of quotient of potentials.
- Many of the hotspots, especially in autumn, hug the borders of Germany. Could this reflect cross-border issues with France, Austria, the Czech Republic? As with (3), the model is more useful to public health officials if it tracks real-world phenomena in an insightful way.
Authors’ Reply: Thank you for pointing this out. This issue is very important, especially in the context of the fact that each country has its own health and pandemic policy. In the Discussion, we added our suggestions and conclusions related to such situations.
Issues with Presentation:
- Please consider re-writing the introduction and conclusion with an eye to brevity and clarity. I read the Introduction 4 times and still did not take away much more than "Researchers have made data-intensive models. We tried to make do with a more minimal more principled model."
Authors’ Reply: Introduction and Discussion have been re-written as suggested by the reviewer.
- You never explicitly define what you mean by independence nor comment how this could relate to the administrative structures or local public health measures instituted by those structures.
Authors’ Reply: As “independence” units we understand units that have a high enough level of infection that they generate new infections themselves, regardless of external influence. We changed the terms in the text to be less ambiguous and added the explanation in the Methodology.
Round 2
Reviewer 2 Report
I thank the authors for carefully revising the manuscript, including bolding important parts, explaining the differences between figures, and succinctly stating that the value of their paper is in showing but not validating a less resource-intensive way to track the dynamics of a spatial distribution, instead of aggregating local indicators of spatial association.
The writing is still not clear enough -- too much passive voice and circumlocutions. But, the text is clear enough to understand now if read twice. Further revisions for clarity are not absolutely necessary.
I reiterate my request to:
- include a paragraph in the discussion that the method, as it currently stands, has no validated predictive ability and
- a paragraph that discusses the application of a validated version of this method. If validated, public health officials might be able to identify regions that require additional interventions, for example. This explicitly addresses the 4th research point, which isn't addressed as explicitly as the others.
Author Response
We would like to thank the reviewer for detailed comments and insightful suggestions. They were invaluable for raising the level of our work. Responses to the comments are provided below.
- Include a paragraph in the discussion that the method, as it currently stands, has no validated predictive ability
Authors’ Reply: A paragraph on method validation has been added in Discussion.
- A paragraph that discusses the application of a validated version of this method. If validated, public health officials might be able to identify regions that require additional interventions, for example. This explicitly addresses the 4th research point, which isn't addressed as explicitly as the others.  
Authors’ Reply: A paragraph that discusses the application of a validated version of the method has been also added.
Additional – small English corrections have been done. Some unnecessary circumlocutions have been deleted.